# Centriole and transition zone structures in photoreceptor cilia revealed by cryo-electron tomography

Zhixian Zhang[1], Abigail R Moye[1,4], Feng He[1], Muyuan Chen[1,2], Melina A Agosto[3], Theodore G Wensel[1]

**Primary cilia mediate sensory signaling in multiple organisms and cell types but have structures adapted for specific roles. Structural defects in them lead to devastating diseases known as ciliopathies in humans. Key to their functions are structures at their base: the basal body, the transition zone, the "Y-shaped links," and the "ciliary necklace." We have used cryo-electron tomography with subtomogram averaging and conventional transmission electron microscopy to elucidate the structures associated with the basal region of the "connecting cilia" of rod outer segments in mouse retina. The longitudinal variations in microtubule (MT) structures and the lumenal scaffold complexes connecting them have been determined, as well as membrane-associated transition zone structures: Y-shaped links connecting MT to the membrane, and ciliary beads connected to them that protrude from the cell surface and form a necklace-like structure. These results represent a clearer structural scaffold onto which molecules identified by genetics, proteomics, and superresolution fluorescence can be placed in our emerging model of photoreceptor sensory cilia.**

## Introduction

Photoreceptors are sensory neurons, which sense and transduce light signals into changes in electrical current and neurotransmitter release. To perform this task robustly, these cells have developed highly ordered compartmentalization and high rates of inter-compartmental protein trafficking. Proteins required for photo-transduction and the structural integrity of the light-sensing outer segment (OS) compartment and the stacks of membranous disks it contains are transported from the inner segment (IS), where biosynthesis primarily occurs, through a thin and crowded intermediate compartment known as the connecting cilium (CC). In each rod cell of the mouse retina, roughly 60 rhodopsin molecules per second are synthesized and packaged into OS membrane disks

(Young, 1967; Williams, 2002; Lyubarsky et al, 2004; Pearring et al, 2013), along with hundreds of other proteins transported to the OS. About 80 new disks are formed per day in each rod. Consequently, this trafficking structure and system must be tightly regulated and well maintained.

The photoreceptor CC and OS together form a unique, specialized sensory organelle with a modified central core templated from the ubiquitous eukaryotic signaling hub, the primary cilium (reviewed in Goldberg et al [2016], May-Simera et al [2017], and Wensel et al [2021]). Primary cilia are cylindrical organelles extending from the cell surface that are involved in extracellular sensing and signaling. The cilia are packed with a distinctive set of lipids and proteins, including receptors, ion channels, microtubule-associated structural proteins, and proteins associated with a complex and tightly regulated bidirectional trafficking machinery (Pazour & Witman, 2003; Berbari et al, 2009; Hilgendorf et al, 2016; Nachury & Mick, 2019; Klena & Pigino, 2022). Cilia generally form after cells exit the cell cycle, as the primary, immotile cilia do in rod-and-cone photoreceptors after their last cell division. Some features of the photoreceptor sensory cilium and its earliest stages of development are similar to those in other ciliated cells, whereas others are unique to photoreceptors (May-Simera et al, 2017; Baehr et al, 2019; Wensel et al, 2021). Ciliogenesis initiates when the centrioles migrate to the apical edge of the inner segment (IS) and form a basal body, consisting of a mother centriole, from which the axoneme eventually grows, and an adjacent daughter centriole, both embedded in a matrix of amorphous appearance but complex organization (Sonnen et al, 2012; Mennella et al, 2014; Fry et al, 2017) known as pericentriolar material. The centrioles contain nine symmetrically arranged microtubule triplets (MTTs) at one end (referred to here as the proximal end), which extend and transition into nine microtubule doublets (MTDs) at the distal end, from which the axoneme grows (Li et al, 2019; Kumar & Reiter, 2021). These MTDs continue throughout the ~1.2 $\mu$m CC, transitioning into singlets as it extends up the ~20-$\mu$m-long OS. At the distal end of the CC (the base of the OS), the light-sensing disks are formed. The region at the base of cilia is often referred to as the transition zone (TZ), and its unique complement of proteins is thought to regulate trafficking through the region, performing a "gate" function (Park & Leroux, 2022). In mouse

[1]Verna and Marrs McLean Department of Biochemistry and Molecular Pharmacology, Baylor College of Medicine, Houston, TX, USA   [2]Division of CryoEM and Bioimaging, SSRL, SLAC National Accelerator Laboratory, Stanford University, Menlo Park, CA, USA   [3]Department of Physiology and Biophysics and Department of Ophthalmology and Visual Sciences, Dalhousie University, Halifax, Canada   [4]Department of Ophthalmic Genetics, Institute of Molecular and Clinical Ophthalmology Basel, Basel, Switzerland

Correspondence: twensel@bcm.edu

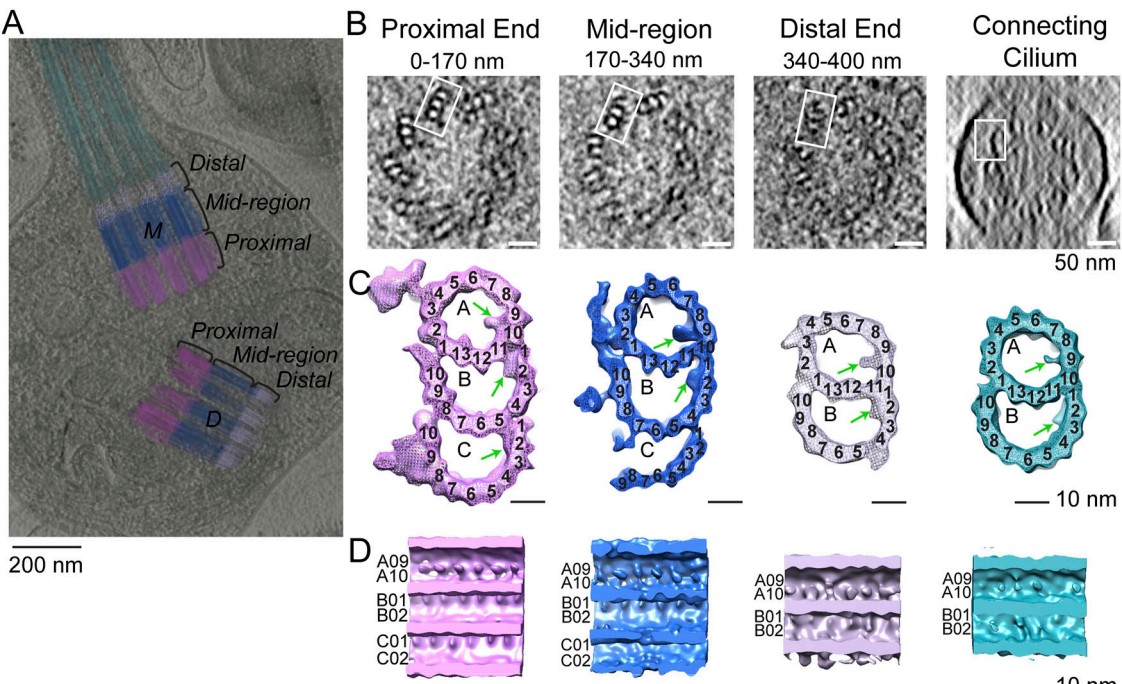

**Figure 1. Tomographic reconstruction and subtomogram averaging of three structural domains from mouse centrioles.**
**(A)** Slice from a reconstructed tomogram showing mother and daughter centrioles (denoted as "M" and "D"). Centrioles were partitioned longitudinally into three regions, the proximal end (0–170 nm, light magenta) consisting of MT triplets, the mid-region (170–340 nm, blue) consisting of incomplete triplets, and the distal end (340–400 nm, light violet) consisting of MT doublets. The extending CC doublets are denoted in cyan. **(B)** Slices of a tomogram of a daughter centriole through the three centriole regions and the CC (from a different part of the tomogram). White boxes indicate examples of approximate regions used for initial stages of subtomogram averaging and refinement. **(C)** Four maps were generated by averaging tomogram subvolumes in each region of centrioles and of the CC. The A-, B-, and C-tubules, including non-tubulin densities, are shown, with protofilament numbering. Microtubule inner proteins are shown with green arrows. **(D)** Maps in C cut in half and rotated 90° toward the viewer to show microtubule inner proteins.

rods, only a subset of TZ proteins are restricted to the base of the CC, with most of those found beyond the triplet-to-doublet transition, whereas others, such as CEP290 (Potter et al, 2021) and SPATA7 (Dharmat et al, 2018), are distributed throughout the entire length of the CC, suggesting the entire CC may contribute to the gate function. In addition, some ciliary proteins such as centrins (Trojan et al, 2008) and EB1 (Hidalgo-de-Quintana et al, 2015) are found in the CC, rather than being restricted to the basal body, as in most other primary cilia.

There are multiple inherited human diseases, known as ciliopathies, because of mutations in genes encoding cilium- or basal body–associated proteins and resulting in defects in cilium structure and function. In many cases, these multi-syndromic diseases include retinal degeneration (Braun & Hildebrandt, 2017; Seo & Datta, 2017; Bachmann-Gagescu & Neuhauss, 2019; Chen et al, 2021), and in a number of cases, retinal degeneration is the only major symptom (Estrada-Cuzcano et al, 2012), highlighting the importance of the BB and CC in photoreceptor function. To better understand how these disorders can cause photoreceptor degeneration, deep knowledge of the structures of these compartments is indispensable.

In this study, we used cryo-electron tomography to determine structures at nanometer-scale resolution of the BB, CC, and associated structures in rod photoreceptors of mice. These findings, and their integration with the growing body of structural, genetic,

and biochemical data on cilia and BB from many other tissues and species (Li et al, 2004; Pazour, 2004; Liu et al, 2007; Dean et al, 2016; Dang et al, 2017; Sun et al, 2019; Danielsson et al, 2020), could help provide insight into the roles of these molecular machines and their components in photoreceptor function and health. Hopefully, these findings will add to the understanding needed for therapies designed to protect or restore visual function in retinal ciliopathies.

# Results

## Structural domains identified from the mouse photoreceptor centrioles and connecting cilium

To facilitate studies of the basal body centrioles (hereafter referred to as centrioles) and connecting cilium, fragments of rods, including rod outer segments (ROS) with attached distal rod inner segments, were isolated and imaged by cryo-electron tomography (cryo-ET) as described previously (Gilliam et al, 2012; Wensel & Gilliam, 2015; Robichaux et al, 2019). The method employed to isolate ROS allows for retention of the entire connecting cilium (CC) and basal body (mother and daughter centrioles). Fig 1A displays a partially segmented 50-nm slice from a reconstructed tomogram, with the mother (M) and daughter (D) centrioles and CC highlighted in

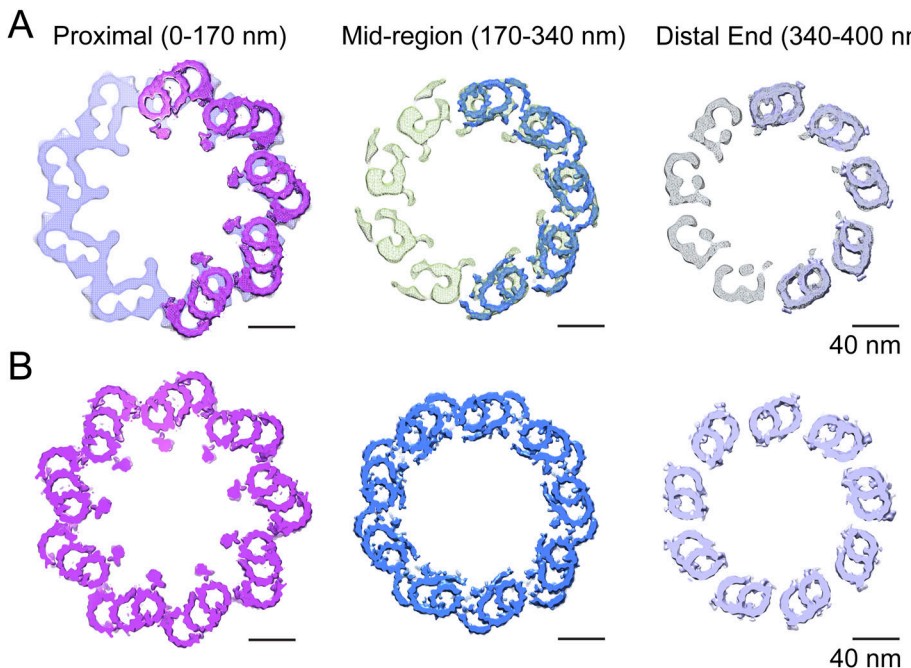

**Figure 2. Centriole model.**
**(A)** Ninefold symmetrized cross sections of a map generated by rotational averaging of a daughter centriole at the proximal (*left*), middle (*middle*), and distal (*right*) regions, respectively, with maps generated by subtomogram averaging of triplet, incomplete triple, or doublet averages (those shown in Fig 1C) fitted onto the symmetrized map and superimposed on the right side of each corresponding map. **(B)** Fully refined, symmetrized, reconstructed/fitted models used for examination of extra-MT densities and inter-MT connections for microtubule triplets and microtubule doublets.

different colors to demonstrate the direction and position of each centriole and CC. The cylinder-shaped basal body has an average length of 400 nm. The proximal region (0–170 nm, light magenta) contains MTTs, the mid-region (170–340 nm, blue) has incomplete triplets transitioning to doublets, and the distal region (340–400 nm, light violet) contains MTDs. Note that the transition to doublets occurs well within the centrioles and not in the CC that emerges beyond the ciliary pocket, as was documented previously in mammalian centrioles (Sun et al, 2019; Le Guennec et al, 2020). The doublets then continue longitudinally into the CC, as shown in cyan in Fig 1A. Cross-sectional slices of the tomograms are shown in Fig 1B, and the corresponding subtomogram average density maps (see details of the averaging below) of the MTTs, incomplete triplets, centriolar MTDs, or CC MTDs are shown in Fig 1C, using the same color scheme. Subtomogram averaging (Fig 1C; see the Discussion section below) allows clear visualization of individual tubulin subunits (13 A-tubule protofilaments, and 10 B- and C-tubule protofilaments, consistent with what is seen in other basal bodies and centrioles [Li et al, 2012; Guichard et al, 2013; Li et al, 2019], as well as non-tubulin proteins both on the inside (microtubule inner proteins, MIPs, Fig 1C and D, discussed below) and on the outside (pinhead, A-C linker, and other external densities) of the MTs.

## Models of the centriole and connecting cilium

In many cases, the cilia and basal body tomogram reconstructions appear elliptical in cross section, rather than being perfectly circular, because of flattening and compression during grid preparation. To improve the signal-to-noise ratio and resolution of the centriolar structure, unflattening and averaging by imposition of ninefold symmetry, as described previously (Robichaux et al, 2019), were applied to selected centriole maps that displayed minimal flattening. Regions of the tomograms containing doublet, triplet, or incomplete triplet MTs were boxed out of raw tomograms in both mother and daughter centrioles and used for subtomogram averaging (Chen et al, 2019) over multiple individual BB and CC. No consistent differences were observed between the mother and daughter centriole maps except for the extended region of doublets forming the CC axoneme extending from the distal end of the mother centriole, so both were used for subtomogram averaging. Separate submaps were used for CC doublets and distal BB doublets. The averaged maps were fit into the symmetrized map of either the centriole or the CC. Fig 2A displays the ninefold symmetry-imposed maps for the proximal, mid-, and distal regions of the BB. Superimposed on the right side of each of these maps are subdomain structures of the MTT, incomplete MTT, or MTD obtained by subtomogram averaging. These were fitted into the symmetry-averaged maps, as shown in the models in Fig 2B, revealing the individual tubulin subunits and non-tubulin structures associated with the centriole. This model, defined in more detail in the Materials and Methods section, was used for further analysis.

## Twist angles observed throughout the basal body and connecting cilium

There are two distinct types of "twist" associated with the arrangement of MTs along the long axis of the centriole and CC. To aid in visualizing these, we fit the average MT maps into raw maps of a centriole (Fig 3) or CC (Fig S1A). One type of twist (Anderson, 1972; Sun et al, 2019) involves the angle of the long axes of the MTDs and MTTs as they wrap helically around the long central axis of the MT bundle. As shown in Fig S1A and B, the outermost (C for triplets, B for doublets) protofilament forms an imperfect right-handed helix with a very long repeat period (>400 nm), such that one period is not

complete along the centriole. The lack of multiple repeats, the variation in this twist angle in the CC versus that in the centriole, the switch from *C* to *B* MTs as the outermost MT during the transition from triplets to doublets, and bending of the CC (Fig 3A) make it difficult to assign a precise number to this period.

A similar gradual twist can be observed in recent reconstructions of bovine motile cilia (Greenan et al, 2020) and primary cilia of epithelial cells (Sun et al, 2019), and was also reported in an early study of monkey oviduct cilia (Anderson, 1972), albeit as a left-handed twist; note that ambiguity in projection direction during reconstruction can lead to flipping of handedness (Anderson, 1972; Chen et al, 2019). A cryo-ET study of primary cilia from MDCK cells revealed a consistent right-handed twist of MT in the basal portion of the cilium of 56.4° ± 7.3° per $\mu$m, which became highly variable in the distal portions (Kiesel et al, 2020). The width of the MT bundle also steadily declines along the proximal-to-distal axis in our maps, ranging in diameter from 230 nm at the proximal end (complete triplet) to 190 nm at the distal end (doublets), because of both the second type of twist described below and the loss of the C MT.

The second type of "twist" previously reported in the literature (Anderson, 1972; Paintrand et al, 1992; Li et al, 2012; Le Guennec et al, 2020), and most readily visualized in cross-sectional views, refers to the angle the row of each MTT or MTD forms with respect to the radius of the 9 + 0 bundle or, equivalently (but with a different value), the angle each triplet or doublet makes with respect to the line connecting it to the adjacent MT row in the bundle (Figs 3C and S1D). We measured this "twist" using two different angles (Figs 3D and S1D). One, the $\alpha$-angle (Greenan et al, 2020), is defined as the outer angle between the lines connecting the centers of A- and B-tubules within each doublet or triplet and the lines connecting the A-tubules of adjacent triplets or doublets, as shown in Fig S1D; and the other, the $\beta$-angle (Li et al, 2012), is defined as the inner angle between the lines connecting the centers of the B-tubules of each doublet or triplet and the line connecting the centers of the A- and B-tubules of the adjacent triplet or doublet (Fig 3C). For ninefold symmetric MT bundles, these two angles have a simple geometric relationship (see the legend of Fig S1), with a nearly constant difference of ~30° for the angles in our maps.

Both angles decrease from the proximal to the distal centriole ends (Figs 3 and S1D). The $\alpha$-angle displays a steady decrease from a 51° MTT angular rotation in the proximal end down to a 25° MTD angular rotation in the CC (Fig S1D). A similar range was observed in mammalian motile cilia (bovine tracheal epithelium [Greenan et al, 2020]). The $\beta$-angle decreased from a 21° MTT angle in the proximal end down to a 6° MTD angular rotation in the distal end of the CC (Fig 3D). Different numbers are obtained depending on whether the raw tomograms or symmetrized maps are used, because of distortions in the angles caused by both physical flattening and computational unflattening (Fig 3E). Measurement of the $\beta$-angle in raw maps of the centriole, without symmetrization, resulted in a 21° angle for the proximal MTTs, ~18° for the incomplete MTTs in the mid-region, and an 11° angular rotation of the MTDs in the distal centrioles (Figs 3E and S1D). These angles differ somewhat when calculated from symmetrized maps (17°, proximal; 13°, mid; and 0°, distal, Fig 3E). Likely, the correct values of these angles in intact retina lie somewhere in between those for symmetrized and non-symmetrized maps, but a definitive answer to this question will

probably depend on obtaining data from samples obtained by high-pressure freezing and focused ion beam milling of more intact retina samples, as has been carried out for other biological samples, including isolated rods (Rigort et al, 2010; Poge et al, 2021; Pinskey et al, 2022; Young & Villa, 2023).

## Non-tubulin complexes associated with centriole microtubules

In addition to the tubulin subunits and protofilaments of the MT, our maps and model have clearly visible features that are formed by proteins other than tubulin isoforms. These include MIPs found within the lumenal regions of the MT and complexes on the outside of the MT such as the pinhead, and the inner scaffold complexes.

### MIPs

Protrusions into the centriolar MT lumen, examples of a general class of MIPs, are seen in the A- (Figs 1, 2, and 4), B-, and C-tubules (Fig 4B), with the one in the A-tubule being most prominent. These are located at protofilaments (PF) A09/A10, B01/B02, and (with the weakest density) C01/02. MIPs have previously been reported in the centriole (Greenan et al, 2018), as well as determination of some of their molecular components (Ichikawa et al, 2017, 2019; Ma et al, 2019; Kiesel et al, 2020; Fabritius et al, 2021; Li et al, 2022).

### Pinhead

One of the conserved structures within the proximal region of the basal body is the pinhead, which protrudes from PF A3 (proto-filament 3 of the A-tubule; see numbering in Fig 4B) into the basal body lumen (Figs 4A and B and S2). The unattached end of the pinhead splits into two densities, or "feet" (PinF1 and PinF2). The longitudinal spacing between each pinhead's PinF2 and the subsequent pinhead's PinF1, which is the same as that between PinF1 and PinF2 of a single pinhead, is 8.4 nm (Fig 4C and D). Although this centriolar structure is conserved throughout species, there have been differences reported in the spacing between pinhead feet, for example, 8.2 nm in CHO cell centrioles (Greenan et al, 2018), and 8 nm in *Chlamydomonas* (Li et al, 2019). There has also been a report in some *Trichonympha* species of a difference in spacing between feet in each pinhead (7.9 nm) as compared to the spacing between adjacent pinheads (8.6 nm) (Nazarov et al, 2020) (Fig S2). The molecular constituents of the pinhead are not known; CEP135 has been proposed as a major component, but a definitive determination has not been made (reviewed in Gönczy and Hatzopoulos [2019]).

### Inter-microtubule connections in the basal body

Connections between adjacent MTT can be seen in our model and are most readily visualized in cross-sectional views (Fig 4E). In the proximal (0–170 nm) region of complete triplets, the A-C linker, a structure connecting the A- and C-tubules in adjacent triplets in this region, can be seen (cross sections, Fig 4F–H; longitudinal views from the outside, 4G, or inside, 4H of the MT bundle). The longitudinal spacing of the A-C linkers is 8.4 nm, as for the pinhead. At the junction between PF C10 and PF B7, there is extra density

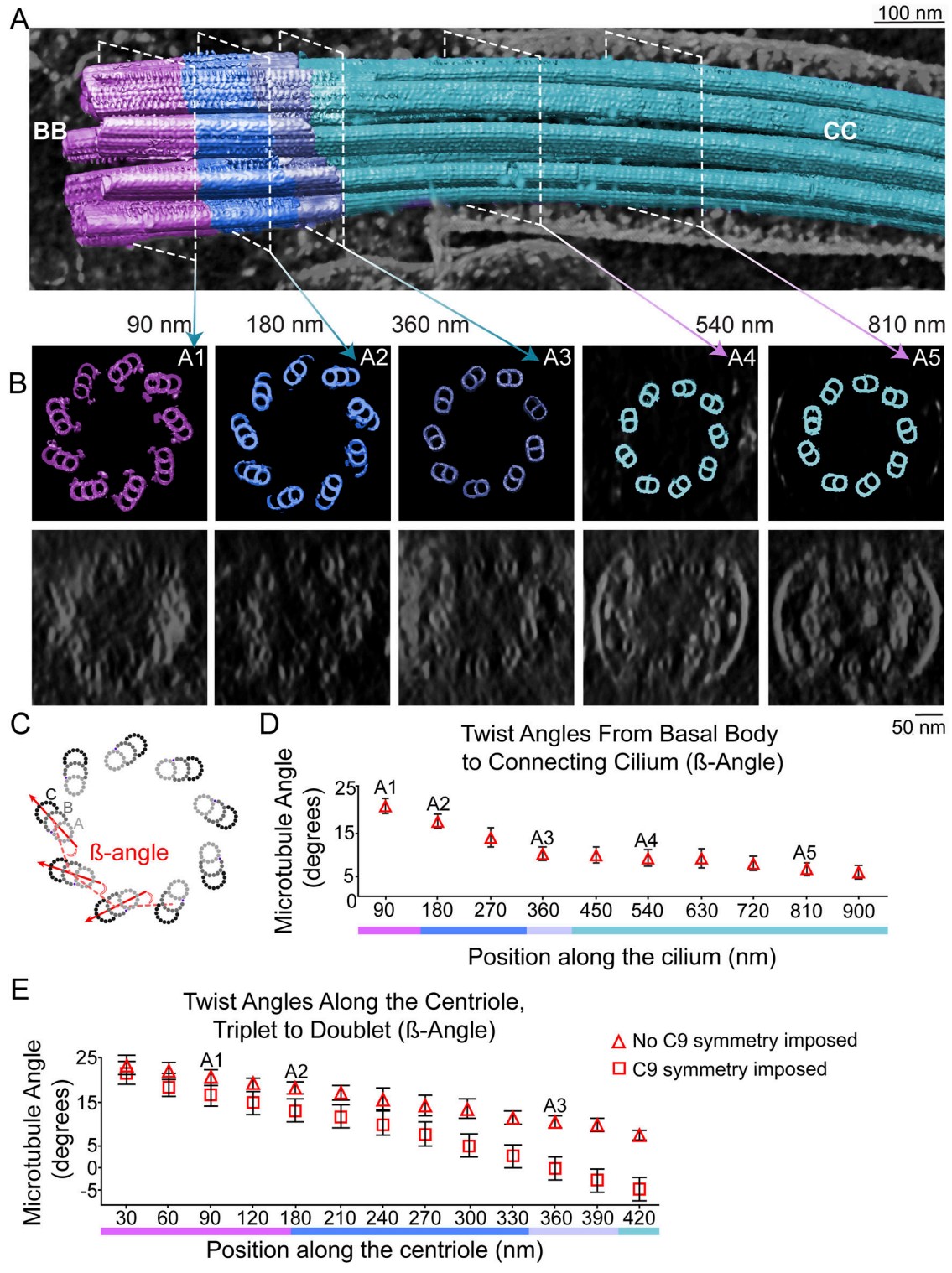

**Figure 3. Fit of MT subtomogram averages into raw tomogram and range of MT twist angles in the cross-sectional plane.**
**(A)** Averaged subvolumes of triplets (cyan; the same subvolumes used for Fig 2) and doublets (magenta) were fitted back into the raw tomogram of an entire mother centriole–CC to generate a whole-cilium map. **(A, B)** Cross sections (top panel, model; bottom panel, raw tomogram) corresponding to the positions indicated by dashed white lines in (A) and denoted by A1–A5 in panel (B). **(C, D, E)** Twist angles $\beta$ (defined in panel (C)) were measured and averaged for each cross section, and are plotted in (D) for the cilium map and (E) for the centriole map as a function of longitudinal position. **(E)** This procedure was repeated using a ninefold rotationally averaged centriole map (Fig S1) and are plotted as boxes in (E). Points and error bars represent means ± SD.

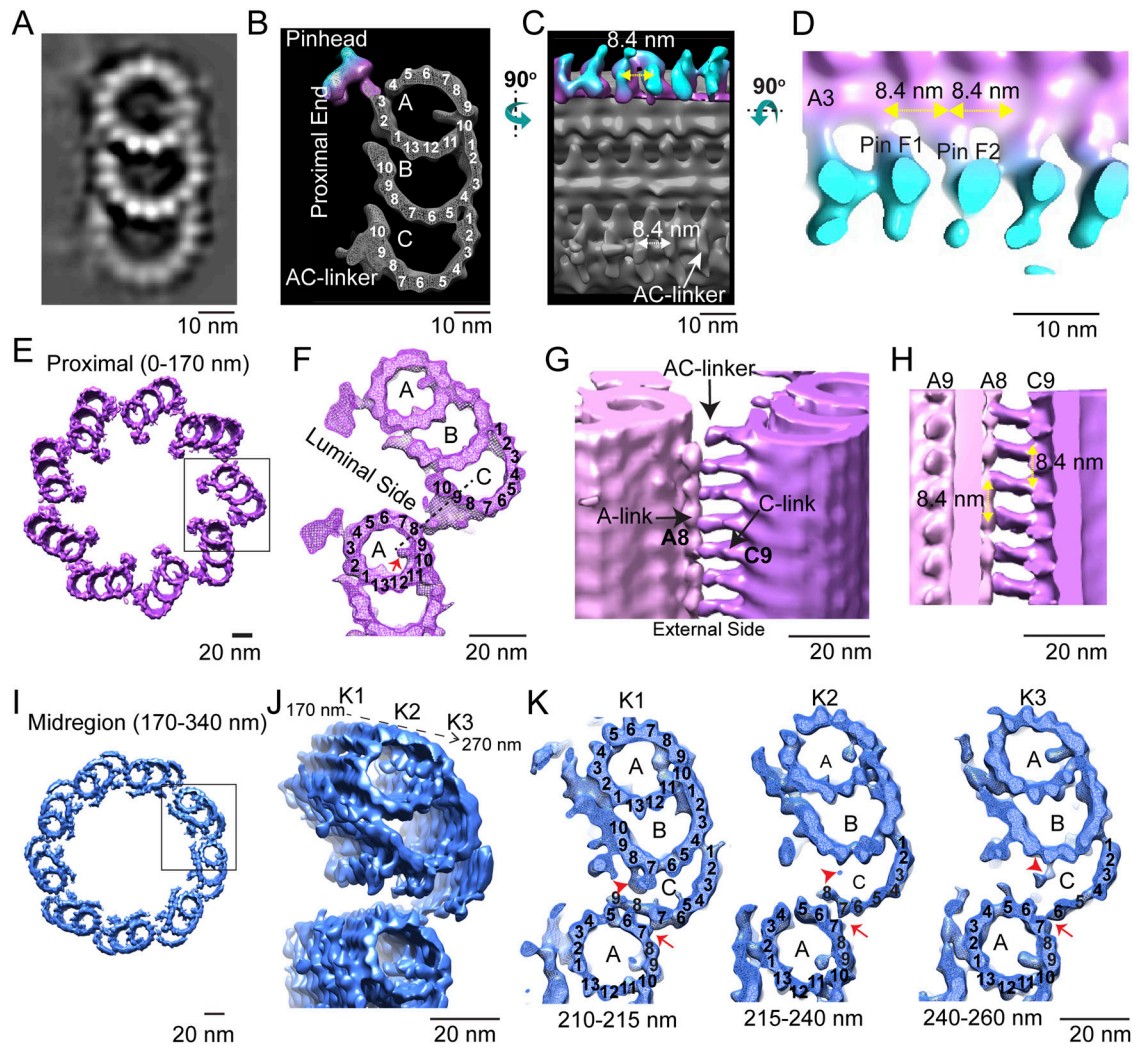

**Figure 4. Pinhead and A-C linker in the proximal and mid-regions of the centriole.**
**(A, B)** Projection view and (B) surface view (with surface capping) of the section through the microtubule triplet (MTT) map obtained from subtomogram averaging of complete MTTs in the proximal region (0–170 nm) viewed down the long axis of the MTT. **(B, C)** Surface rendering from (B) rotated 90° to reveal the side facing the lumen of the MT bundle. The pinheads are colored with depth-coding (magenta closer to MT, cyan furthest from the MT surface). **(C, D)** is a magnified and rotated version of the rendering in (C), highlighting the PinF1 and PinF2 feet of the pinhead and their 8.4-nm spacing. **(E)** Surface rendering of the centriole model in the proximal region of the mother centriole (0–170 nm) (from Fig 2B). **(E, F)** Magnified view of the boxed area in (E). The A-C linker is the density that links PF A8 on the A MT at the bottom to PF C9 of the adjacent C MT of the MTT at the top. **(G)** Side view from the outside of the MT bundle highlighting the A-C linkers. **(F, H)** Cutaway view (the cutting plane is shown as a dotted line in (F)) from the lumen of the MT bundle with yellow arrows highlighting the 8.4-nm spacing of the A-C linkers. **(I, J, K)** Mid-region (170–340 nm) of the compete centriole model showing the incomplete MTTs; the proximal 100 nm of the boxed region is shown at a higher magnification in (J). **(J)** is a tilted version showing the progression of structural changes along the centriole axis. **(J, K)** shows cross-sectional views of three different subregions (K1–K3) extracted at different levels from the map as shown in (J). Red arrows and arrowheads indicate the variations in the connections between A MT and C MT along the axis.

associated with C10 (Fig 4), which may be a non-tubulin subunit. There is a gradual change in these inter-MT connections in the mid-region (170–340 nm, Fig 4I; tilted view, Fig 4J) of incomplete triplets. The gradual loss of the C MTs, first of PF C9 and C10, and eventually the remaining C PFs is apparent. There is a persistent, albeit weak, density associated with PF B7, near the original position of C10, which may represent the non-tubulin protein at partial occupancy (Fig 4K, red arrowhead). The gradual loss of the C MT is accompanied by a replacement of the A-C linker connection with what appear to be direct connections of C9–A6 and of C7–A7. In the progression from triplets through incomplete triplets to doublets, there is first

(210–215 nm) a break at the inner junction of the C-tubule. The A-C linker is remodeled, and in its place, the major attachments observed are initially between PF A5, A6, and A7 of the A-tubule, and C9, C8, and C7, respectively (Fig 4K, panel K1). As the C-tubule becomes further disassembled, the connections to the neighboring A-tubule are lost, with the C6–A7 connection being the last to remain (Fig 4K, panels K2, 215–240 nm, and K3, 240–260 nm).

The transitions of connections between the A-tubule and C-tubule along the distance from the proximal to the distal ends have also been observed in centrioles or basal bodies from other species and cell types with either motile or sensory cilia. In the CHO

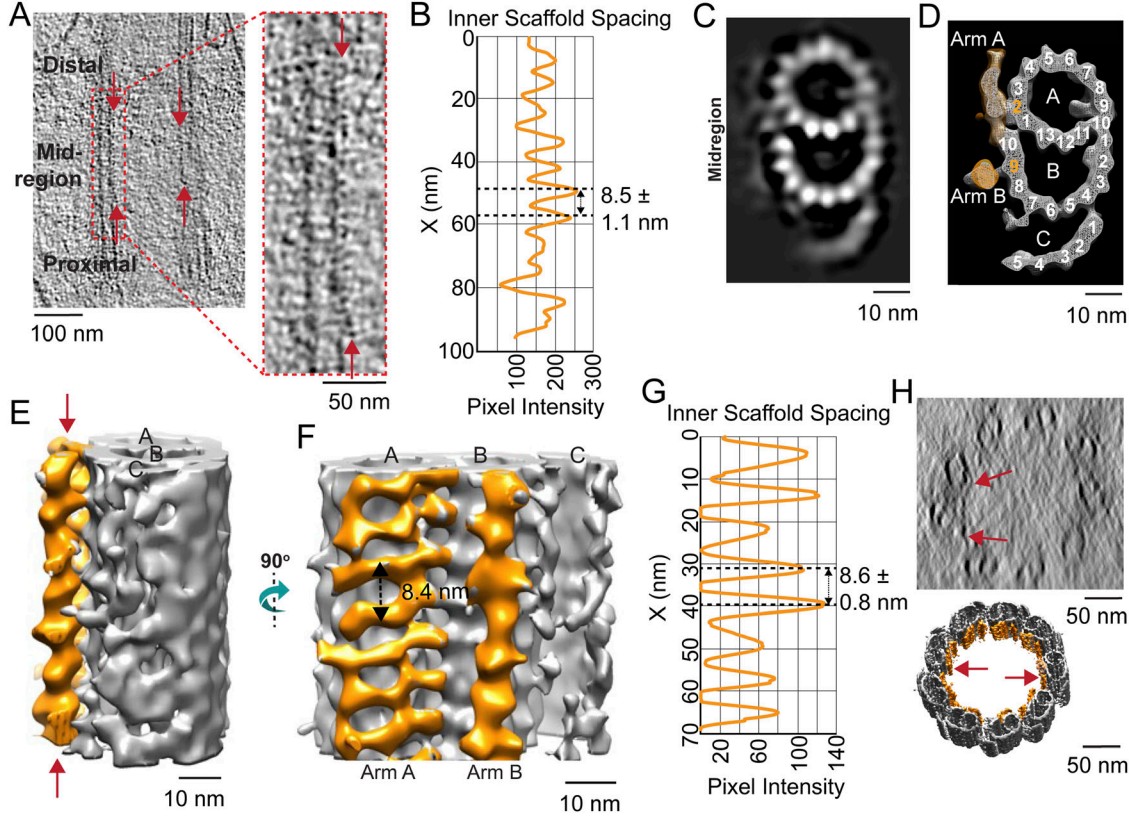

**Figure 5.  Inner scaffold along the incomplete triplets in the mid-region of the basal body.**
**(A)** Section from raw cryo-electron tomogram of the rod photoreceptor showing the basal body structure in a longitudinal view with labels for the proximal, mid-region, and distal positions along the BB. The red arrows around the mid-region indicate the visible inner scaffold densities attached to the MTs. A zoomed-in view of inner scaffold densities attached to microtubule triplets (as indicated with a dashed red box) is shown to the right. **(B)** Corresponding line intensity profile (right) of the inner scaffold showing an average periodicity in the mid-region of 8.5 nm. **(B, C, D)** Cross sections of the BB mid-region map from subtomogram averaging in a projection view (B) or a surface view with surface capping (D). These reveal the complete A- and B-, and partial C-tubules. In panel (D), the extending filamentous densities from the lumen side of PF A2 (Arm A) and PF B9 (Arm B) are highlighted in orange. **(C, D, E, F)** Longitudinal views of the same region as in (C, D), with the inner scaffold shown in *orange* bracketed by red arrows in (E). **(E)** View in (F) is of the map shown in (E) rotated 90° toward the viewer. **(F, G)** Line intensity profile along the direction indicated by the double-headed arrow in (F) is shown in panel (G) (periodicity of 8.6 nm). **(H)** Cross-sectional view of a section of a raw tomogram in the BB mid-region with red arrows indicating the presence of inner scaffold (top) and a slightly tilted view of the corresponding region of the model from Fig 2B highlighting the inner scaffold indicated in *orange* (bottom).

cell centriole, at the proximal A-C linker, PF A9 is connected to C8/C9, and at the distal end, PF A9 is directly bound to PF C8 (Greenan et al, 2018). In *Chlamydomonas reinhardtii,* at the proximal end, PF A6 is connected to the neighboring triplet at the tail end of the C-tubule, and at the distal end of the BB, PF A6 is connected to PF C7/C8 (Li et al, 2012). It is unclear whether the proteins that make up the A-C linker change throughout this transition region, or whether it is the same proteins that have remodeled to allow detachment and eventual loss of the C-tubule.

### Inner scaffold structure in the basal body

Along the lumenal side of the MT bundle, the presence of filamentous structures attached to the A- and B-tubule, termed Arm A and Arm B (Fig 5), was observed throughout the mid-region (170–340 nm) and the distal region (340–400 nm) of the centriole (Fig 5A, arrows). Together, these form the structure termed the inner scaffold. At the boundary between the proximal triplets and mid-region incomplete triplets, there is density in the Arm A region

whose position is similar to some of the density observed in the pinhead. It is unclear whether distinct molecular components give rise to these densities. Line intensity plots from the raw tomograms demonstrated that these filamentous densities exhibited 8.5 ± 1.1 nm spacing along the length of the microtubules (Figs 5B and S3). Longitudinal sections (Fig S4A) and cross sections (Fig S4B) of rotationally averaged maps (Fig S4C) also clearly show this inner scaffold in the mid- and distal centriole regions, as well as in the CC, compared with the proximal centriole region (Fig S4A–C). Whether or not the proteins that form this inner scaffold change from centriole to CC is unclear. To enhance the structural detail for the inner scaffold, subtomogram averaging of the structure was carried out, followed by back-fitting into the centriole map from Fig 2B. This reconstruction allowed for clear visualization of the two distinct densities: Arm A (attached to A2) and Arm B (attached to B9) (Fig 5C–F, indicated in orange). Both have a periodicity of 8.4–8.6 nm (Fig 5B, F, and G), similar to spacings in other cell types (Fig S5A–C). In cross sections from the 3D reconstructions, these densities formed an almost continuous ring (Fig 5H, lower panel), even though only

partial inner scaffold filaments were observed in the raw tomogram because of sample compression and the missing wedge in tilt-series images of individual centrioles (Fig 5H, upper panel). An additional density between Arm A and Arm B seen in some other centrioles (a "stem" in, e.g., *P. tetraurelia* [Le Guennec et al, 2020]) was not apparent in our maps (Fig S5D and E). In centrioles from other species and cell types (Le Guennec et al, 2020), this inner scaffold has been consistently observed, first appearing in the incomplete triplet region and extending over various distal lengths (Fig S5B–E).

### Ciliary necklace beads and bridges to MT observed along the connecting cilium

Two of the hallmarks of the transition zone region at the base of cilia are (1) the presence of arrays of external protuberances from the ciliary membrane, known as the "ciliary necklace" (Gilula & Satir, 1972), and (2) connections between the MT and ciliary membrane known as "Y-shaped links" or "Y-links" because of their greater width at the membrane as compared to the narrow neck connected to the MT. Photoreceptor CC were among the first cilia identified as having a ciliary necklace, through freeze-fracture studies that implicated their association with intramembranous particles (Rohlich, 1975), observed throughout most of the length of the CC. Photoreceptor CC also have Y-links not only at their base, but also throughout their length (Potter et al, 2021); however, neither the relationship between the Y-links and necklace beads nor their three-dimensional structure has been clearly determined, and their molecular compositions are also unknown.

Examples of the appearance of the ciliary necklace can be seen in raw tomograms (Fig 6A) and in conventional transmission electron microscopy (TEM) images (Fig 6C). Ciliary necklace beads encircle the cilium as strands or rows along the circumference of the ciliary membrane, lying roughly in planes tilted slightly from those perpendicular to the ciliary axis, as seen in the surface rendering of the 3D map (Fig 6D). The intramembranous strands are roughly parallel to one another, and many, if not all, appear to span only a portion of the circumference of the CC membrane.

Measurements of the longitudinal distances between adjacent beads along the length of the CC (Fig 6B) revealed that the beads were ~40 nm apart along most of the CC, whereas closer to the basal body (ciliary neck), the beads were ~22 nm apart (n = 7, Fig 6B). The 40-nm distance corresponds well to the spacing of intramembranous particles seen in a previous freeze-fracture study of rat rods (Rohlich, 1975). Our measurements using conventional TEM images of ultrathin sections of mouse CC yielded an average bead spacing along the CC of ~35 nm (not shown), consistent with previous measurements from such images of ~32 nm in rat rods (Besharse et al, 1985). The differences could reflect distortions arising from the extensive processing of the conventional TEM and freeze-fracture/SEM samples, as compared to the unfixed nature of the flash-frozen cryo-ET samples.

Volumes containing beads of the ciliary necklace were extracted and subjected to subtomogram averaging, followed by fitting of the average structure into the raw tomogram of a CC base (see the Materials and Methods section). In the resulting map (Fig 6E), the ciliary necklace beads appear to contain two major domains

connected within the ciliary membrane. An outer bead structure, ~40 nm in the circumferential dimension (purple inverted-head arrows, Fig 6D) and <10 nm in the longitudinal dimension (pink arrows, Fig 6D), gives rise to the "necklace" appearance. An inner bridge is composed of multiple small subdomains, suggestive of a noisy average of heterogeneous structures filling the distance between the MT and the membrane (Fig 6D, right panel). The circumferential span of the outer bead structure appears to have five globular ridges, which are clearly distinguished in conventional TEM images (Fig 6E and F) and in an averaged subtomogram map of such images (Fig 6G) but are barely resolved in the raw tomograms (Fig 6A). The distance between these five ridges differs slightly between subtomogram average maps (~11 nm, Fig 6G and H) and conventional TEM images (~10 nm, Fig 6F and H), and is close to the measurements described from freeze-fracture studies of rat rods (Rohlich, 1975).

### Y-link structures along the connecting cilium

When maps of MTDs with connected partial inner bridges/links and of beads are fitted into a ninefold rotationally averaged CC map (Fig 7A), prepared by computational straightening, unflattening, and rotational averaging of a single CC map, as described previously (Robichaux et al, 2019), it is apparent that there is good alignment and partial overlap of the two structures (Fig 7B). Within the CC, a narrow neck-like structure is connected at a somewhat variable position between PF A10 and PF B1, which extends and widens in the radial direction toward the ciliary membrane (Fig 7A and B). The structure of this Y-link appears to be somewhat heterogeneous, as is apparent in the variations between different longitudinal regions in the rotationally averaged maps (Fig 7E) and in conventional TEM images (Fig 7C and F). Because of this heterogeneity, the inner bridge/link portion of the "bead" map obtained by subtomogram averaging likely represents an average of multiple distinct structures rather than an accurate representation of a single structure. Despite the intrinsic differences in sample preparation referred to above, the cryo-ET and conventional TEM results are remarkably well aligned (Figs 3, 7C and D). From both TEM and cryo-ET results, the widest part of the inner stalk (Y-link) was measured to be ~41 nm. Integration of the results from raw tomograms, sub-tomogram averages, conventional TEM, and prior literature leads to the model illustrated in Fig 7G. In this model, the wide portion of the Y-links occupies a ~15° arc of the CC circumference, so there is a 25° separation between adjacent Y-links, and there is continuous density leading from the MT doublet to and through the ciliary membrane, forming five globular ridges per Y-link.

## Discussion

### Conserved and unique structural features of photoreceptor ciliary centrioles

The overall architecture of the centrioles associated with the photoreceptor sensory cilium is remarkably similar to those associated with other cilia, including both motile and primary cilia,

**Figure 6. Subtomogram averaging of the intramembranous beads/ciliary necklace along the ciliary membrane.**

**(A)** Sections from raw tomograms from mouse rod photoreceptors displaying intramembranous beads periodically arranged on the ciliary plasma membrane (left displaying a slice from the outside view of the CC; right showing a slice through the center of the CC). The beads were categorized into two groups based on spacing, ciliary pocket region (orange arrows, ~22-nm spacing), and CC region (purple arrows, ~40 nm). **(B)** Spacing between beads for each of the two groups along the ciliary membrane (~22 nm, n = 6 cells for ciliary pocket; and ~40 nm, n = 7 cells for CC). The box-and-whisker plot shows the median (horizontal line), upper and lower quartile (box edges), and range of measurements (error bars), as well as individual points. **(C)** Conventional electron micrographs of mouse rod CC displaying the same intramembranous beads (average spacing 38.5 nm apart in this image, magenta arrows) along the CC. The dashed-line box in the image on the far left indicates the region shown at a higher magnification in the (middle) image immediately to its right. Dashed-line boxes on the image to the far right are the regions shown at a higher magnification below. **(D)** Map of the ciliary necklace beads obtained by subtomogram averaging was fit (in multiple copies) into a raw tomogram of a CC and the resulting map shown in a surface view from outside the CC (top, far left), in a projection view sliced through the CC membrane (immediately to the right of the surface view), and a magnified surface view (lower left). The series of images on the right show different high-magnification views of the average bead (cyan) and the portion of the CC membrane into which it was fit (gray). The longest dimension of the averaged structure is about 40 nm (magenta bars with inverted arrowheads), and the narrower dimension is about 10 nm (pink

and including those of protists/metazoan organisms. However, many of the structural details we have observed can vary substantially among cell types and developmental stages and have not previously been determined for mammalian photoreceptors. Some of these details suggest rods may differ in multiple ways from other mammalian sensory cilia, even in the regions proximal to the highly structurally divergent outer segment.

### Longitudinal microtubule twisting

The variation in MTT angle in the cross-sectional plane along the centriole axis is well documented in multiple ciliated cells. The helical twist of the MT is perhaps less well appreciated, albeit also well documented. Reviews of ciliary or centriolar structures tend to feature cartoons showing perfectly straight MT in both structures, absolutely parallel to the long axes. The functional significance of this twist is unclear, but it has been proposed (Sun et al, 2019) to help sensory cilia withstand mechanical stress because of flow. It is hard to imagine a need for this function in photoreceptors, as the close-packed nature of outer segments holds the cilia fairly rigidly in place. However, as the results of our method of preparing cell fragments for microscopy demonstrate, the photoreceptor CC and basal body are remarkably resistant to such forces. In those experiments, the outer segments are broken off from the retina by shear forces, but the CC and BB consistently remain intact, so a stabilization function for the twisting cannot be ruled out.

### Width and cross-sectional twist angle shifts in BB in other cell types

Although the general features of the BB structure are conserved across a wide range of species and cell types, there are many examples of dramatic divergence (reviewed in Gomes Pereira et al [2021], Jana [2021], Kumar and Reiter [2021], and LeGuennec et al [2021]). A number of centrioles from various species have been studied using electron tomography under cryogenic conditions (e.g., Guichard et al, 2010; Li et al, 2012, 2019; Guichard & Gönczy, 2016; Greenan et al, 2018; Le Guennec et al, 2020). A change in diameter from ~230 to 220 nm from the triplet to the incomplete triplet was previously observed in a cryo-ET study of CHO cell centrioles (Greenan et al, 2018), but in that case, the incomplete triplet was reported to be located in the distal centriole instead of in the middle as observed here. In some studies on mammalian centrioles, twist angle variations along the axis of up to 25° have been observed, but in contrast to what we observe in the murine photoreceptor BB, the diameter of the basal body barrel was reported to be constant along its length, unless treated with EDTA (Anderson, 1972; Paintrand et al, 1992), suggesting a role of divalent cations. In basal bodies of motile cilia of *C. reinhardtii*, the centriole barrel was observed to have a constant value of 260 nm throughout its length (Li et al, 2012). However, in a more recent study comparing multiple species centrioles (Le Guennec et al, 2020), the diameter was seen to change from proximal to distal, and in *C. reinhardtii*, the diameter was seen to be the greatest in the mid-region of the centrioles. These diameter variations could be due to the triplet-to-doublet transition, changes in twist angles, and/or the change from the A-C linker to the inner scaffold.

### Non-tubulin complexes: MIPs and inter-MT connections

We observe clear densities corresponding to non-tubulin proteins in the lumen of the centriole MT. Although these have been observed in multiple cell types, their positions in motile cilia appear to vary from those in photoreceptors. In most cell types, the molecular identity of these proteins is unknown. The A09/10 MIP has been observed in other centrioles (Greenan et al, 2020) and has been identified in motile ciliary axonemes (Ichikawa et al, 2017; Li et al, 2022). This MIP is well conserved throughout species and is seen in our centriole maps, as well as the maps we have produced of the A-tubule in the CC. The proteins CFAP53, CFAP161, SPAG8, NME7, CFAP141, and CFAP95, which are found at the A09/A10 seam, are thought to be this MIP (Gui et al, 2021). The C01/C02-bound MIPs were not observed in the triplets with the incomplete C-tubule. The proteins RIB72A and RIB72B have been identified as components of MIPs in the A-tubule in motile cilia of protists (Stoddard et al, 2018). The related mammalian protein EFHC1 (EF-hand–containing protein 1)/myoclonin is a microtubule-associated protein identified as a ciliopathy protein linked to myoclonic epilepsy and present in motile cilia; however, it is not present in mammalian neurons (Suzuki et al, 2020). There is no current information on the proteins making up the MIPs in the photoreceptor BB, though it is likely they are highly conserved because of (1) the similar shape and placement of the MIPs observed between previous mammalian maps and the photoreceptor centriole, and (2) the fact that mutations in many of these MIPs cause ciliopathies, which are often associated with blinding disease, for example, CFAP20 (Chrystal et al, 2022).

Also uncertain are the identities of the proteins extending into the lumen of the MT bundle, such as the pinhead and those making up the inter-MT connections, the A-C linker, and the inner scaffold. CEP135 has been proposed as a major component of the pinhead, but a definitive determination has not been made (reviewed in Gönczy and Hatzopoulos [2019]). The inner scaffold has been described in centrioles from multiple species and has been hypothesized to act as the glue that maintains centriole cohesion and strength under compressive forces (Le Guennec et al, 2020). It has been proposed that the proteins that make up this inner scaffold structure, based on expansion microscopy in human cell lines, include POC1B, FAM161A, POC5, and centrin-2 (Le Guennec et al,

---

double-headed arrows). **(E, F)** Conventional transmission electron microscopy (TEM) images of CC showing the beads. **(E, F)** Dashed-line box in the upper image of (E) shows the region shown at higher magnifications below and in panel (F), to the right. The magenta bar with two inverted arrowheads represents 40 nm, and the cyan-outlined arrowheads indicate the ridge-like subdomains of the structure that yield the "bead" appearance. **(D, G)** is a similar view of a portion of a map from cryo-ET obtained as described in panel (D) above. **(H)** Plot of spacings measured for bead ridges in our TEM images and subtomogram averages, or in a previously published report of freeze-fracture/SEM studies. Whisker plots show medians, lower and upper quartiles, and range of the data, with n = 5 cells for both TEM and subtomogram average results.

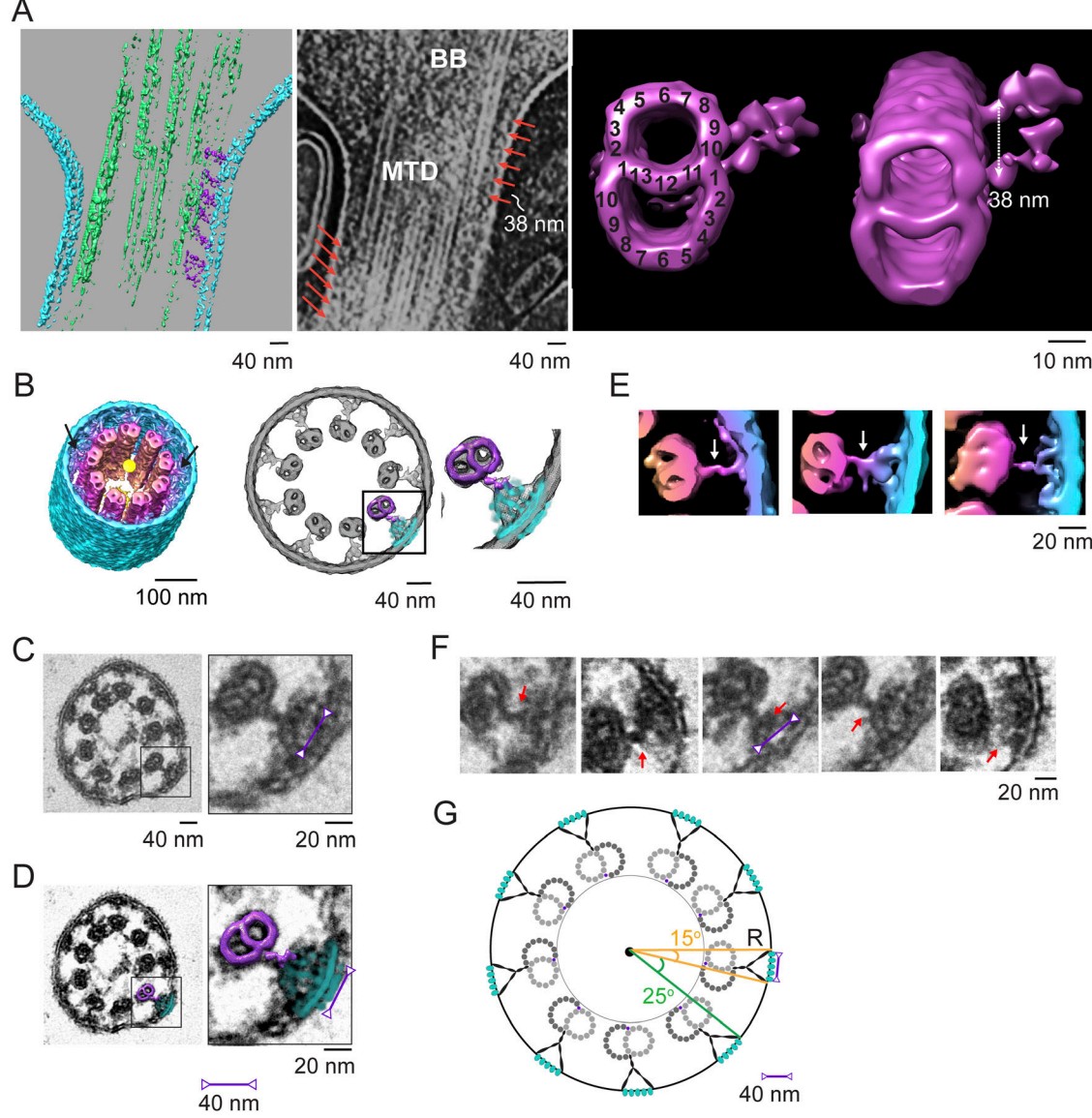

**Figure 7. Y-link connections to microtubules.**
**(A)** *Left panel*, a segmented version; and *middle panel*, a solid representation of the tomogram used to generate the template for the subtomogram average of doublets with attached bridges/links (*right panel, magenta*). BB, basal body, MTD, microtubule doublet. The MT-to-membrane bridges/links are indicated with orange arrows. The subtomogram average map on the right displays longitudinal spacings of 38 nm (*double-headed arrow*), also seen in the raw tomogram (*orange arrows*). The attached bridging density emerging from the A/B junction (A10/B1) is consistent with the observation in the CC longitudinal view (*left* and *middle panels*). **(B)** Rotationally averaged map of the CC shown in tilted (left) and cross-sectional (center) views; arrows point to Y-links. The averaged doublet + filamentous density map (magenta) and the averaged ciliary bead map (from Fig 6, cyan) were fit into the rotationally averaged map of the CC (gray mesh), providing a composite model of the MTD–Y-link–membrane complex (a square box in the *center* panel and an expanded view on the *right*). For comparison, in (C), a conventional transmission electron microscopy (TEM) image of a cross-sectional view of a detergent-extracted CC is shown at two different magnifications, with a magenta double-headed arrow displaying a distance of 41 nm. **(B, D)** MTD–Y-link ciliary model, aligned as in panel (B), is shown superimposed on the TEM image in (D). **(B, E)** View of the same map as in (B), with the cross section taken at a different position along the axis, and with the distance from the center color-coded. Three individual MTD–Y-link–membrane–bead complexes at different axial positions are shown to the right, with white arrows indicating the bridging/link density. **(F)** Different cross-sectional images from conventional TEM of CC, with red arrows pointing to the filamentous densities. **(G)** Schematic model of the MTD–Y-link–membrane–bead complexes in the CC cross section. **(E, F)** "Y" portion is meant to denote the approximate position, rather than the three-dimensional structure, given the heterogeneity demonstrated in (E, F).

2020). Because mutations in these proteins are associated with retinal degenerative diseases in humans (https://web.sph.uth.edu/RetNet/disease.htm#10.205d) or animal models (Ying et al, 2019; Mercey et al, 2022), it is possible they form the inner scaffold of centrioles in photoreceptors as well. This structure also resembles the inner scaffold circle observed in conventional TEM throughout the length of the CC. Recently, POC5, centrin, and FAM161A were reported to be localized within the MTs of photoreceptor CC (Mercey et al, 2022), and centrins have long been known to localize within the MTs along the length of photoreceptor

CC (Wolfrum & Salisbury, 1998; Robichaux et al, 2019; Potter et al, 2021), consistent with these proteins participating in the inner scaffold. Further work will need to be done to examine where exactly these proteins localize and to determine the structures of the associated protein complexes. An additional non-tubulin complex is the "inner junction" found adjacent to PF10 (Linck et al, 2014) of MT B. This complex has recently been identified as containing the proteins FAP126, FAP106, and FAP276 in motile cilia from protists (Khalifa et al, 2020), and was previously reported to be made up of PACRG and CFAP20 (Dymek et al, 2019). Defining the proteins that directly interact with these at the "stem" of Arm A would be of great interest.

### Limitations of our maps and model

Here, we report three distinct compartments in the centrioles of the photoreceptor cilium, with the transition to MTDs occurring within the centriole, providing the template for extension of the doublet-containing axoneme. There is a substantial mid-region with incomplete MTT, but the transition to doublets is clearly complete within the distal centriole, well proximal to the base of the axoneme. It is not clear whether this is the general case for mammalian sensory cilia or a special feature of a subset, including those in photoreceptors. We have not included distal or subdistal appendages in our models, which have been built from subtomogram average maps of limited volumes and are based primarily on the MT. These structures, which have been extensively studied in other cell types (Bowler et al, 2019; Chang et al, 2023), are visible in our maps, but would require much larger volumes in the boxing step carried out before subtomogram averaging.

### New insights into Y-links and ciliary necklace

The Y-link–containing CC is close to 1 $\mu$m in length, considerably longer than typical ciliary transition zones. Although our subtomogram average of this structure is quite noisy, our data provide new insights into the common features of what appears to be a somewhat heterogeneous structure, which appears at positions corresponding to the ninefold positions of the MTD. One of the features that is common among the diverse Y-links within our samples is their connections to the transmembrane portion of the ciliary necklace beads, which appear to be less heterogeneous in structure, but somewhat more irregular in their placement on the surface of the ciliary membranes. With subtomogram averaging and the increased number of Y-links and ciliary necklace beads in each CC as compared to other transition zones, we were able to, for the first time, resolve these structures to 30–38 Å resolution, and to provide strong support to the long-proposed hypothesis that these form one large complex structure extending from the MT bundle through the membrane to its surface. The protein composition of these structures remains uncertain, and it is not clear what their functional roles or dynamical properties are. Combining electron microscopic techniques with superresolution fluorescence, proteomics and genetics should clarify these questions in the future.

# Materials and Methods

### Animals

C57BL/6 WT mice, aged 4–6 wk, were used for this study. All procedures were approved by the Baylor College of Medicine Institutional Animal Care and Use Committee and adhered to the Association of Research in Vision and Ophthalmology guidelines for the humane treatment and ethical use of animals for vision research.

### ROS purification

All mice were maintained in a 12/12-h light (400 Lux)/dark cycle. WT C57BL/6J mice were purchased from Jackson Lab (Bar Harbor, ME). 1-mo-old C57BL/6J mice used for tissue samples were euthanized by $CO_2$ inhalation before dissection following the American Association for Laboratory Animal Science protocols.

Preparation of purified mouse ROS was modified from Wensel and Gilliam (2015). Briefly, to isolate ROS with CC and portions of the inner segment attached, retinas were dissected under dim red light and placed in 200 $\mu$l Ringer's buffer (10 mM Hepes, 130 mM NaCl, 3.6 mM KCl, 1.2 mM MgCl$_2$, 1.2 mM CaCl$_2$, and 0.02 mM EDTA, pH 7.4) with 8% (vol/vol) OptiPrep (iodixanol; Sigma-Aldrich). Retinas were pipetted up and down with a 200-$\mu$l-wide orifice tip 50 times and then centrifuged at 400$g$ for 2 min at RT. The supernatants containing ROS were collected. The process was repeated four to five times. All ROS were pooled and loaded onto the top of a gradient of 10%, 15%, 20%, 25%, and 30% (vol/vol) OptiPrep step-gradient, and centrifuged for 60 min at 19,210$g$ at 4°C in a TLS-55 rotor (Beckman Coulter). The ROS band was collected with an 18G needle, diluted with Ringer's buffer to 3 ml, and pelleted in a TLS-55 rotor for 30 min at 59,825$g$ at 4°C. The ROS pellet was resuspended in Ringer's buffer for cryo-electron microscopy.

### Cryo-ET

Isolated ROS were processed for cryo-ET as described previously (Gilliam et al, 2012; Wensel & Gilliam, 2015; Robichaux et al, 2019). Briefly, isolated ROS were mixed with BSA-stabilized 15 nm fiducial gold (2:1), and 2.5–3 $\mu$l of the mixture was deposited on freshly glow-discharged 3.5/1 200-mesh Quantifoil carbon-coated holey grids. Samples were allowed to settle for 15 s before blotting from either the front or back side and plunge-frozen in liquid ethane using a Vitrobot Mark III automated plunge-freezing device. The frozen/hydrated sample was imaged on a Polara G2 electron microscope (FEI Company), equipped with a field emission gun, operated at 300 kV using a direct electron detector camera (Gatan K2 Summit).

Single-tilt image series were automatically collected using SerialEM (Mastronarde, 2005) software at a defocus range of 8–10 $\mu$m and a magnification of 9,400 × (equivalent to 4.5 Å/pixel). The total electron dose per tomogram was ~50–70 e/Å$^2$ for 35 tilt images, covering an angular range of −51° to +51° with 3° increments (±51°, 3° increment). At each tilt, angle "movies" of ~8 frames were collected and MotionCorr (Li et al, 2013) was used to correct the image drift within each tilt before merging them. Alignment and

3D reconstruction were performed automatically using the work-flow in EMAN2 software (Chen et al, 2019).

## Subtomogram averaging and correspondence analysis

Structural domains within centrioles and connecting cilia (CC) were visualized and examined longitudinally using IMOD software (Kremer et al, 1996; Mastronarde & Held, 2017). Centriole reconstructions were divided into three regions: the proximal region (0–170 nm), containing complete triplets with intact C-tubules; the mid-region (170–340 nm), containing incomplete triplets with partial C-tubules; and the distal region (340–400 nm), containing doublets. The CC (400 nm–CC), containing doublets, was analyzed separately. Subvolumes from these regions were extracted and assigned to different groups. Subsequently, subtomogram averaging was performed for each group using the EMAN2 package (Chen et al, 2019).

From 25 centrioles (16 tomograms), ~1,800 subvolumes from the proximal region, ~1,600 subvolumes from the middle region, and ~400 subvolumes from the distal region, with dimensions of 100 × 100 × 100 nm, were boxed and extracted. The starting models within each group (low-pass-filtered to 60 Å) were first generated using EMAN2 without applying any symmetry, followed by iterative subtomogram refinements performed by EMAN2 using the corresponding starting models. Resolutions for the refined maps were determined using the gold-standard method of splitting particles into two groups and measuring their Fourier shell correlation 0.143 criterion. For the refinement of triplets, we also used the average triplet structures from CHO centrioles (proximal, EMD-7776; distal, EMD-7777 [Greenan et al, 2018]) as starting models. After a few iterations of refinement, the two different approaches resulted in the consistent/nearly identical maps for both triplets. The final averaged structures have a resolution of 33 Å for both triplets and 40 Å for the doublet. For the doublet averages from CC, ~1800 subvolumes with a dimension of 100 × 100 × 100 nm were boxed out and yielded a final averaged map at ~30 Å resolution after several iterative subtomogram refinements by EMAN2.

To further investigate the structure of doublets with "Y-links," ~1,600 subvolumes of a MTD with membrane-directed densities along the CC were boxed and extracted out with a dimension of 100 × 100 × 100 nm. After several iterative subtomogram refinements using the average doublet structure from CC as the initial model, the final model map achieved ~30 Å resolution. The resulting map had a short density attached to the MTD, but could not be unambiguously interpreted with regard to the orientation (luminal versus membrane side of MT).

To help identify the luminal versus membrane side of doublets with Y-links, a single tomogram with longitudinal Y-link spacings of ~38 nm was selected (Fig 7A). 150 subvolumes of MTDs with membrane-directed densities along the CC were boxed out with a dimension of 100 × 100 × 100 nm (including three 32-nm repeats). After several rounds of iterative refinement using EMAN2, a map with ~38-nm Y-link spacing was obtained, in which the Y-links with 32-nm spacing were clearly visible on the membrane-directed side (Fig 7A). This map was used for fitting into the rotationally symmetrized CC map as shown in Fig 7.

For the average of ciliary necklace beads, ~400 subvolumes with a dimension of 50 × 50 × 50 nm were boxed along the plasma membrane of ciliary neck and CC regions. After the starting model and iterative refinement were performed by EMAN2, the final map achieved a resolution of ~38 Å.

Ninefold averaged maps were used for MTT/doublet angular twist analysis and "Y-link" visualization in the CC. The centrioles and the cilia with the least distortion of the diameter were picked and visualized by IMOD and Chimera, and ninefold symmetry average was applied in EMAN2 as previously described (Robichaux et al, 2019). To increase the contrast, some of the maps (Figs 3E and 4G–H and S1B) are presented in a 4xbinned format with a voxel size of 17.8 Å in EMAN2.

## Model building and visualization

The centriole models from the proximal to the distal end as shown in the figures (Figs 2, 4, and 7, S1B, and S5D) were built by fitting averaged triplets and doublets manually into a ninefold symmetrized map in UCSF Chimera (Pettersen et al, 2004) and then optimizing the fitting using the built-in "fit in map" tool. The models for the whole cilia (BB + CC) (Fig 3) were built by repeatedly fitting the averaged triplets (containing 3-tubulin heterodimers in length) and the doublets back into a raw tomogram. The resulting maps were displayed/presented in a 4xbinned format in EMAN2. Fitting of averaged structures of MTD with attached Y-links and of ciliary necklace beads was carried out with both raw tomograms and ninefold averaged maps. All fitting and superpositions of ciliary necklace beads in TEM images and subtomogram averages, as well as tomographic reconstructions and 3D surface rendering of subtomogram averages, were generated and visualized using IMOD (Mastronarde & Held, 2017) and UCSF Chimera (Pettersen et al, 2004) (http://www.rbvi.ucsf.edu/chimera).

## Ambient temperature transmission electron microscopy

Mice were euthanized under deep anesthesia by transcardial perfusion with fixative (2% PFA, 2% glutaraldehyde, and 3.4 mM $CaCl_2$ in 0.2 M Hepes, pH 7.4). Eyes were enucleated, the cornea and lens were removed, and the eyecups were placed in fixative (2% PFA, 2% glutaraldehyde, and 3.4 mM $CaCl_2$ in 0.2 M Hepes, pH 7.4) for 2 h, rocking at RT. The eyecups were prepared using a similar protocol to that described previously (Potter et al, 2021). Briefly, the eyecups were embedded in 4% agarose from which 150-$\mu$m vibratome sections were cut. These sections were subsequently stained (rocking at RT) with 1% tannic acid/0.5% saponin in 0.1 M Hepes, pH 7.4, for 1 h, followed by 1% uranyl acetate in 0.2 M maleate buffer, pH 6.0, for 1 h. The sections were dehydrated in a series of 15-min ethanol washes (50%, 70%, 90%, 100%, and 100%), followed by infiltration with Ultra Bed Epoxy Resin (Electron Microscopy Sciences). The sections were embedded in resin between two ACLAR sheets sandwiched between glass slides in a 60°C oven for 48 h. At 24 h, the top slide and ACLAR sheet were removed, and resin blocks in BEEM capsules (Electron Microscopy Sciences) were stamped onto each section to allow for polymerization for the next 24 h.

For increased resolution of the Y-links, mice were euthanized by $CO_2$ asphyxiation followed by cervical dislocation, and after

enucleation, retinas were dissociated from eyecups and incubated rolling in 0.1% Triton X-100 in 1xPBS for 1 h at 4°C. After rinsing with 1xPBS, they were fixed for 2 h in the same fixative as above. The retinas were directly stained, dehydrated, and infiltrated as above in half-dram (1.35 ml) glass vials. They were embedded in BEEM flat embedding molds in a 60°C oven for 48 h.

Ultrathin sections (50–70 nm) were cut on a Leica UC7 ultramicrotome and were placed on copper grids, and poststained in 1.2% uranyl acetate in Milli-Q water for 6 min, followed by staining in Sato's lead (a solution of 1% lead acetate, 1% lead nitrate, and 1% lead citrate; all from Electron Microscopy Sciences) for 2 min. Grids were imaged either on a JEOL 1400 Plus electron microscope with an AMT XR-16 mid-mount 16-megapixel digital camera or on a JEOL JEM-1400Flash 120-kV TEM with a high-contrast pole piece and a 15-megapixel AMT NanoSprint15 sCMOS camera. For each microscope, AMT software was used for image acquisition and images were subsequently cropped with slight contrast adjustments in FIJI//ImageJ (Schneider et al, 2012).

### Twist angle measurements

For the estimation of twist angle, two methods were used. In the first method ($\beta$-angle), nonagon vertices were manually placed in the center of each B-tubule and a midline drawn through the centers of the tubules of each triplet/doublet, and then, the angle between each triplet/doublet (by midline) and its closest nonagon edge was measured. This method was used for the angular estimation in centriole maps (BB) with and without imposition of ninefold symmetry, and also for the whole-cilium raw map (BB + CC). In the second method ($\alpha$-angle), nonagon vertices were placed in the center of A-tubules rather than B-tubules. This method was used only for the angular estimation along the whole-cilium length (BB + CC) for comparison with motile cilia (Greenan et al, 2020). The angles were measured and averaged from the cross section every 90 nm and displayed as a function of distance throughout the cilium axis (proximal–distal end).

## Data Availability

Cryo-ET data and calculated maps have been deposited with the Electron Microscopy Data Bank under EMDB entry IDs EMD-41812, EMD-41813, and EMD-41814. All other data are available on request (twensel@bcm.edu).

## Supplementary Information

## Acknowledgements

Funding was provided by grants from the NIH (R01-EY026545, R01-EY031949, F32-EY031574, and R21-MH125285) and the Welch Foundation (Q0035). Thanks to the BCM cryo-EM core, the cryo-EM core at the University of Texas Health Sciences Center, Houston, and Lita Duraine of the TEM core at the Neurological Research Institute, Houston.

## Author Contributions

Z Zhang: formal analysis, investigation, visualization, and writing—review and editing.
AR Moye: formal analysis, investigation, visualization, and writing—original draft, review, and editing.
F He: resources, investigation, and writing—review and editing.
M Chen: software, formal analysis, and methodology.
MA Agosto: formal analysis, visualization, and writing—review and editing.
TG Wensel: conceptualization, formal analysis, supervision, funding acquisition, validation, project administration, and writing—review and editing.

## Conflict of Interest Statement

The authors declare that they have no conflict of interest.

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
