## [Reviewer comments · Life Science Alliance]

Life Science Alliance

Centriole and transition zone structures in photoreceptor cilia revealed by cryoelectron tomography

Zhixian Zhang, Abigail Moye, Feng He, Muyuan Chen, Melina Agosto, and Theodore Wensel

DOI: <https://doi.org/10.26508/lsa.202302409>

Corresponding author(s): Theodore Wensel, Baylor College of Medicine

Review Timeline:

Submission Date:	2023-09-30
Editorial Decision:	2023-11-27
Revision Received:	2023-12-05
Editorial Decision:	2023-12-07
Revision Received:	2023-12-12
Accepted:	2023-12-12

Transaction Report:

November 27, 2023

Re: Life Science Alliance manuscript #LSA-2023-02409-T

Dr. Theodore G Wensel
Baylor College of Medicine
Biochemistry and Molecular Biology
One Baylor Plaza
Houston, Texas 77030-3411

Dear Dr. Wensel,

Thank you for submitting your manuscript entitled "Centriole and transition zone structures in photoreceptor cilia revealed by cryoelectron tomography" to Life Science Alliance. The manuscript was assessed by expert reviewers, whose comments are appended to this letter. We invite you to submit a revised manuscript addressing the Reviewer comments.

Thank you for this interesting contribution to Life Science Alliance. We are looking forward to receiving your revised manuscript.

Sincerely,

B. MANUSCRIPT ORGANIZATION AND FORMATTING:

Reviewer #1 (Comments to the Authors (Required)):

This is an excellent work describing the structural features of the base of the rod photoreceptor cilia. CryoET images of high quality show the arrangement of Y-shaped links and the ciliary necklace. The manuscript is well-written and was a pleasure to read. As described in the abstract, the structure obtained provides a scaffold for future studies that will identify proteins/molecules that are part of the cilia. The major claim is the clarity of the structural features provided, validation with TEM, twisting observed at the base, and organization of the ciliary necklace in rod photoreceptors. While the functional significance of the structures is unclear at this moment, the study provides a solid foundation upon which the field could build. One could argue that additional animal models are needed to validate the findings, but the approach is tedious and should be part of a separate study. Overall, the findings are of great interest to anyone interested in photoreceptor biology or ciliopathies.

Reviewer #3 (Comments to the Authors (Required)):

The submitted manuscript by Zhang and colleagues titled, "Centriole and transition zone structures in photoreceptor cilia revealed by cryo-electron tomography", provides the first in-depth cryoET analysis of the mouse rod photoreceptor cilium. They include analysis of the microtubule structure and associated protein densities found within the centrioles of the basal body and connecting cilium region of the axoneme. The authors discuss microtubule twisting, TMT to DMT transition, inner scaffold proteins, Y-links, and the ciliary necklace. This is the first study to analyze at the structure of the ciliary necklace. Their subtomogram averaging finds that the ciliary necklace is not bead-like but consists of a rectangular density that is composed of 5 ridges. This is a very interesting finding and the authors do well explaining how this structure is mis-represented single dense membrane particle in standard TEM images. Where this study falls short is in the shallowness of the cryoET data that prevents higher resolution structures to be resolved. It would have been a step above if the authors were able to localize or map known protein structures back onto the densities (even tubulin subunits). Without that information many of their conclusions feel vague. Words such as "unknown", "undetermined", "unclear" show up throughout the results when describing different densities or regions. One concern, is that despite this uncertainty, the authors are bold enough to state in the discussion that many of the current models need to be adjusted based on their data. Without identifying molecular complexes, I feel that the authors are overstating the power of their analysis. However, this study is the best structural analysis performed on photoreceptor cilia to date and is therefore an important advancement that should be considered for publication by Life Science Alliance. A list of major and minor remarks regarding the writing can be found below:

Major Remarks:

In the introduction the authors state "The region at the base of cilia where the microtubules undergo a transition from triplets to doublets is often referred to as the transition zone (TZ)". However, in the results the authors state "Note that the transition to doublets occurs well within the centrioles..." There is a disconnect between these two statements. If the transition to doublets is occurring within the mother centriole of the basal body, then why state that the TZ is marked by the transition from triplets to doublets? It is well described in the cilia field that the TZ is part of the axoneme structure. Even the citation the authors list in the introduction, Park and Leroux 2022, states that the "The first segment of the axoneme, termed transition zone (abbreviated TZ), contains typically Y-shaped structures, termed Y-links, that physically connect the doublet microtubules to the overlying ciliary membrane." The authors desire to attribute the TZ to the transition between triplets to doublets is unfounded, confusing, and should be removed.

Figure 6B lists n=6 and n=7 for the box and whisker plot. Does this n= measurements or cells or biological replicates?

Figure 6H does not provide n's for the plot.

In the first discussion section the authors start by saying the photoreceptor cilium is highly conserved, but also suggest current models should be adapted to include the differences they identified. It is not clear how the differences they observed would alter any of the current models. The authors need to discuss the major observed differences in detail, so this makes more sense.

In the second discussion section ends abruptly. Are the authors implying that the longitudinal twist of the MTs is providing structural integrity to the outer segment? If so, that needs to be clearly stated.

Minor Remarks:

Reference formatting error identified in the Introduction - References listed as (1,2) and (3) cannot be identified in the references as this list is alphabetical.

The use of abbreviations is not consistent throughout the manuscript. Microtubule (MT) is missing from the abbreviation list. I am confused by the references listed as follows "high-pressure freezing and focused ion beam milling of more intact retina samples (Poge et al., 2021; Rigort et al., 2010; Young and Villa, 2023; Zhao et al., 2021)." These references did not perform high-pressure freezing and focused ion beam milling on intact retina samples.

Figure 3 Legend typo. Extra period should be removed "The twist angles β (defined in panel C) were measured and averaged for each cross-section and plotted in D for the cilium map and E for the centriole map. as a function of longitudinal position."

THEODORE G. WENSEL, PH.D.
Robert A. Welch Professor
Verna and Marrs McLean Department of
Biochemistry and Molecular
Pharmacology
One Baylor Plaza
MS: BCM125
Houston, TX 77030-3411
713-798-4528 office
713-796-9438 FAX

Dear Dr. Sawey,

Thank you for your message of Nov. 27, 2023 concerning our manuscript, "Centriole and transition zone structures in photoreceptor cilia revealed by cryoelectron tomography," by Zhixian Zhang et al. We have revised the manuscript in response to the reviewers' very useful and constructive comments. Our point-by-point response to those comments and the revisions made in response are listed below. The reviewers' comments are displayed in black Arial 11 point font and **our responses are displayed in red 12 point Calibri font.**

Reviewer #1 (Comments to the Authors (Required)):

This is an excellent work describing the structural features of the base of the rod photoreceptor cilia. CryoET images of high quality show the arrangement of Y-shaped links and the ciliary necklace. The manuscript is well-written and was a pleasure to read. As described in the abstract, the structure obtained provides a scaffold for future studies that will identify proteins/molecules that are part of the cilia. The major claim is the clarity of the structural features provided, validation with TEM, twisting observed at the base, and organization of the ciliary necklace in rod photoreceptors. While the functional significance of the structures is unclear at this moment, the study provides a solid foundation upon which the field could build. One could argue that additional animal models are needed to validate the findings, but the approach is tedious and should be part of a separate study. Overall, the findings are of great interest to anyone interested in photoreceptor biology or ciliopathies.

We thank the reviewer for their positive comments and agree that additional studies with animal models are needed. Such studies are currently ongoing in our laboratory but will take some time to complete.

Reviewer #3 (Comments to the Authors (Required)):

The submitted manuscript by Zhang and colleagues titled, "Centriole and transition zone structures in photoreceptor cilia revealed by cryo-electron tomography", provides the first in-depth cryoET analysis of the mouse rod photoreceptor cilium. They include analysis of the microtubule structure and associated protein densities found within the centrioles of the basal body and connecting cilium region of the axoneme. The authors discuss microtubule twisting, TMT to DMT transition, inner scaffold proteins, Y-links, and the ciliary necklace. This is the first study to analyze at the structure of the ciliary necklace. Their subtomogram averaging finds that the ciliary necklace is not bead-like but consists of a rectangular density that is composed of 5

ridges. This is a very interesting finding and the authors do well explaining how this structure is mis-represented single dense membrane particle in standard TEM images. Where this study falls short is in the shallowness of the cryoET data that prevents higher resolution structures to be resolved. It would have been a step above if the authors were able to localize or map known protein structures back onto the densities (even tubulin subunits). Without that information many of their conclusions feel vague. Words such as "unknown", "undetermined", "unclear" show up throughout the results when describing different densities or regions. **We agree that fitting known structures into our maps would have increased the information content; however, at our current resolution, while we could, for example, fit tubulin subunits into the maps, the resulting high-resolution pseudo maps would not be accurate, despite having the appearance of high resolution and molecular certainty. Performing such fits reliably will depend on improving further the resolution obtained *via* subtomogram averaging, and finding ways to identify unambiguously specific non-microtubule proteins.** One concern, is that despite this uncertainty, the authors are **bold enough to state in the discussion that many of the current models need to be adjusted based on their data.** Without identifying molecular complexes, I feel that the authors are overstating the power of their analysis. **We have revised the discussion to avoid making such claims. The concluding sentence of the first paragraph of the discussion has been revised to read, "Some of these details suggest rods may differ in multiple ways from other mammalian sensory cilia, even in the regions proximal to the structurally highly divergent outer segment," and does not mention existing models.** However, this study is the best structural analysis performed on photoreceptor cilia to date and is therefore an important advancement that should be considered for publication by Life Science Alliance. A list of major and minor remarks regarding the writing can be found below:

Major Remarks:

In the introduction the authors state "The region at the base of cilia where the microtubules undergo a transition from triplets to doublets is often referred to as the transition zone (TZ)". However, in the results the authors state "Note that the transition to doublets occurs well within the centrioles..." There is a disconnect between these two statements. If the transition to doublets is occurring within the mother centriole of the basal body, then why state that the TZ is marked by the transition from triplets to doublets? It is well described in the cilia field that the TZ is part of the axoneme structure. Even the citation the authors list in the introduction, Park and Leroux 2022, states that the "The first segment of the axoneme, termed transition zone (abbreviated TZ), contains typically Y-shaped structures, termed Y-links, that physically connect the doublet microtubules to the overlying ciliary membrane." The authors desire to attribute the TZ to the transition between triplets to doublets is unfounded, confusing, and should be removed.

We have removed the text, "where the microtubules undergo a transition from triplets to doublets," so the sentence now reads, "The region at the base of cilia is often referred to as the transition zone (TZ)".

Figure 6B lists n=6 and n=7 for the box and whisker plot. Does this n= measurements or cells or biological replicates? **It is now made clear in the figure legend that n = cells**

Figure 6H does not provide n's for the plot. **These are now provided in the figure legend**
In the first discussion section the authors start by saying the photoreceptor cilium is highly conserved, but also suggest current models should be adapted to include the differences they identified. It is not clear how the differences they observed would alter any of the current

models. The authors need to discuss the major observed differences in detail, so this makes more sense.

The text about adapting models has been removed to avoid confusion (see the revised text above).

In the second discussion section ends abruptly. Are the authors implying that the longitudinal twist of the MTs is providing structural integrity to the outer segment? If so, that needs to be clearly stated.

The following text has been added, "so a stabilization function for the twisting cannot be ruled out."

Minor Remarks:

Reference formatting error identified in the Introduction - References listed as (1,2) and (3) cannot be identified in the references as this list is alphabetical. The references have been fixed.

The use of abbreviations is not consistent throughout the manuscript. Microtubule (MT) is missing from the abbreviation list. The use of abbreviations has been made consistent and "MT" as well as other abbreviations have been added to the list

I am confused by the references listed as follows "high-pressure freezing and focused ion beam milling of more intact retina samples (Poge et al., 2021; Rigort et al., 2010; Young and Villa, 2023; Zhao et al., 2021)." These references did not perform high-pressure freezing and focused ion beam milling on intact retina samples. The text has been altered to read, "high-pressure freezing and focused ion beam milling of more intact retina samples, as has been carried out for cryo-ET on other biological samples, including isolated rods (Pinsky et al., 2022; Poge et al., 2021; Rigort et al., 2010; Young and Villa, 2023)."

Figure 3 Legend typo. Extra period should be removed The extra period has been removed."The twist angles β (defined in panel C) were measured and averaged for each cross-section and plotted in D for the cilium map and E for the centriole map. as a function of longitudinal position."

[end of reviewers' comments]

The revised manuscript and original figures have all been uploaded. We hope the revised paper will be found suitable for publication in Life Science Alliance.

Sincerely,

Theodore G. Wensel
Baylor College of Medicine
Welch Professor Department of Biochemistry and Molecular Pharmacology
Professor, Departments of Neuroscience and Ophthalmology
Director, Houston Area Molecular Biophysics Program

December 7, 2023

RE: Life Science Alliance Manuscript #LSA-2023-02409-TR

Dr. Theodore G Wensel
Baylor College of Medicine
Biochemistry and Molecular Pharmacology
One Baylor Plaza
Houston, Texas 77030-3411

Dear Dr. Wensel,

Thank you for submitting your revised manuscript entitled "Centriole and transition zone structures in photoreceptor cilia revealed by cryoelectron tomography". We would be happy to publish your paper in Life Science Alliance pending final revisions necessary to meet our formatting guidelines.

- please add your main, supplementary figure, and table legends to the main manuscript text after the references section
- please add a conflict of interest statement to your main manuscript text
- please use the [10 author names et al.] format in your references (i.e., limit the author names to the first 10)
- there are call outs for figures S6, S7A and these figures are not provided -- please correct
- please add callouts for Figures 7C; S1A-B; S3A-C; S4A-C; S5A-E to your main manuscript text

Figure Checks:

-In Figure 6C, the resolution of the bottom right panel appears less than the other panels. If this can be improved, please do so.

A. FINAL FILES:

B. MANUSCRIPT ORGANIZATION AND FORMATTING:

Sincerely,

December 12, 2023

RE: Life Science Alliance Manuscript #LSA-2023-02409-TRR

Dr. Theodore G Wensel
Baylor College of Medicine
Biochemistry and Molecular Pharmacology
One Baylor Plaza
Houston, Texas 77030-3411

Dear Dr. Wensel,

Thank you for submitting your Research Article entitled "Centriole and transition zone structures in photoreceptor cilia revealed by cryoelectron tomography". It is a pleasure to let you know that your manuscript is now accepted for publication in Life Science Alliance. Congratulations on this interesting work.

DISTRIBUTION OF MATERIALS:

Again, congratulations on a very nice paper. I hope you found the review process to be constructive and are pleased with how the manuscript was handled editorially. We look forward to future exciting submissions from your lab.

Sincerely,
